# Regulation of Ras Signaling by S-Nitrosylation

**DOI:** 10.3390/antiox12081562

**Published:** 2023-08-04

**Authors:** Sónia Simão, Rafaela Ribeiro Agostinho, Antonio Martínez-Ruiz, Inês Maria Araújo

**Affiliations:** 1Algarve Biomedical Center Research Institute (ABC-RI), University of Algarve, 8005-139 Faro, Portugal; rragostinho@ualg.pt; 2Faculty of Medicine and Biomedical Sciences, University of Algarve, 8005-139 Faro, Portugal; 3Unidad de Investigación, Hospital Universitario Santa Cristina, Instituto de Investigación Sanitaria Princesa, 28009 Madrid, Spain; amartinezruiz@salud.madrid.org; 4Departamento de Bioquímica y Biología Molecular, Facultad de Farmacia, Universidad Complutense de Madrid, 28040 Madrid, Spain; 5Champalimaud Research Program, 1400-038 Lisbon, Portugal

**Keywords:** Ras, nitric oxide, S-nitrosylation, post-translational modifications, neuronal cells

## Abstract

Ras are a family of small GTPases that function as signal transduction mediators and are involved in cell proliferation, migration, differentiation and survival. The significance of Ras is further evidenced by the fact that Ras genes are among the most mutated oncogenes in different types of cancers. After translation, Ras proteins can be targets of post-translational modifications (PTM), which can alter the intracellular dynamics of the protein. In this review, we will focus on how S-nitrosylation of Ras affects the way these proteins interact with membranes, its cellular localization, and its activity. S-Nitrosylation occurs when a nitrosyl moiety of nitric oxide (NO) is covalently attached to a thiol group of a cysteine residue in a target protein. In Ras, the conserved Cys118 is the most surface-exposed Cys and the preferable residue for NO action, leading to the initiation of transduction events. Ras transduces the mitogen-activated protein kinases (MAPK), the phosphoinositide-3 kinase (PI3K) and the RalGEF cellular pathways. S-Nitrosylation of elements of the RalGEF cascade remains to be identified. On the contrary, it is well established that several components of the MAPK and PI3K pathways, as well as different proteins associated with these cascades, can be modified by S-nitrosylation. Overall, this review presents a better understanding of Ras S-nitrosylation, increasing the knowledge on the dynamics of these proteins in the presence of NO and the underlying implications in cellular signaling.

## 1. Introduction 

Ras are signal transduction proteins involved in cell proliferation, migration, differentiation and survival. Ras genes are also proto-oncogenes and the most frequently mutated genes in cancer, accounting for 30% of human cancers. In this context, Ras proteins have been described in the last years as attractive drug targets; however, due to the lack of a drug-binding pocket on their surface, they are referred to as undruggable entities [1,2]. Presently, several promising strategies are being developed to circumvent this fact, based either on targeting upstream molecules or downstream effectors in the signaling of Ras cascade [2,3]. After translation, Ras can undergo post-translational modifications (PTM), which modify the structure of the protein, the way Ras interacts with membranes and therefore the underlying signaling pathways [1,4,5]. S-Nitrosylation is a PTM of cysteine residues, which can modify Ras subcellular localization and activity, with several implications for cellular physiology [6,7]. In the present work, we review the S-nitrosylation of Ras and its implications in cellular signaling, with a particular focus in neuronal cells. Recently, we described S-nitrosylation of Ras as an important mechanism for the proliferation of neural stem cells (NSC) in an in vivo model of brain lesion [6,8,9,10], pointing to the significance of this PTM in the context of neurogenesis. Presently, the challenge is to clarify what are the functional effects and identify the physiological role of this modification in neurons, so novel approaches directed to Ras can be defined. Very recently, Nakamura and colleagues precisely reviewed how S-nitrosylation impacts protein misfolding in the context of neurodegenerative diseases, highlighting the important role that this PTM has in the brain [11].

## 2. Ras Biology

Three genes code for four Ras isoforms (H-Ras, K-RasA and K-RasB splice variants, and N-Ras). Ras are 21 kDa proteins with 188–189 amino acid residues (188 amino acids in the case of K-Ras4B and 189 for all other Ras isoforms). These proteins share about 90% total sequency identity and all display two key domains: the catalytic domain (G domain) containing residues 1–166 and the hypervariable region (HVR) containing residues 167–189. The amino acid sequence of the HVR is flexible and distinctive among the four Ras proteins and contains the C-terminal CAAX motif (C, cysteine; A, aliphatic amino acid; X, any amino acid). The CAAX motif is isoform specific. CVLS refers to H-Ras, CVVM refers to N-Ras, CIIM refers to K-Ras4A and CVIM refers to K-Ras4B [1,5,12,13,14]. 

Ras proteins are small GTPases that use guanine nucleotide binding for cellular signaling. Ras localizes in the inner leaflet of the plasma membrane and is activated upon GTP binding, following upstream activation of tyrosine kinase membrane receptors, and inactivated after GTP is hydrolyzed to GDP. The conversion between the active and inactive state of Ras proteins is controlled by regulatory proteins: the GTPase accelerating proteins (GAPs) and the guanine nucleotide exchange factors (GEFs). GAPs stimulate the GTPase activity and promote the hydrolysis of GTP. GEFs are responsible for GDP dissociation and for promoting the formation of nucleotide-free Ras that then can bind a GTP molecule [1,5,14,15] (Figure 1). Ras activated by GTP binding, possibly followed by dimerization, couples to cellular effectors, such as Raf, mitogen-activated protein kinase (MAPK) or phosphoinositide-3 kinase (PI3K), and many others, through a Ras-binding domain (RBD) in the target effector [12,16]. 

## 3. Post-Translational Modifications of Ras Are Major Regulators of Its Function and Subcellular Localization

Several PTM of Ras occur in the CAAX motif present in the C-terminal domain of these proteins. Prenylation, the first PTM needed for Ras to become active, is a covalent modification that assists protein attachment to cellular membranes. This PTM refers to the addition of a specific carbon chain to proteins. For instances, a 15-carbon (farnesyl) or 20-carbon (geranylgeranyl) isoprenoid moieties can bind to the cysteine present at the C-terminal of a protein through a thioether linkage [12,17]. Specifically in the case of Ras, the addition of a farnesyl group fosters Ras interaction with membranes and is catalyzed by farnesyltransferase. This reaction occurs at Cys186 for all Ras isoforms, excepts for K-Ras4B where it occurs at Cys185 [12,14].

Subsequently, the modified Ras travels to the endoplasmic reticulum where a methylation reaction (transfer of a methyl group to a protein) occurs at the C-terminal Cys of the CAAX motif upon the proteolytical deletion of the AAX tripeptide [5,18]. The isoforms N-Ras, H-Ras and K-Ras4A are then directed to the Golgi and palmitoylated on its Cys residues of the HVR, before they can target the plasma membrane. Palmitoylation is a PTM involving the addition of a 16-carbon fatty acid (palmitic acid) to a cysteine residue of a protein via a thioester bond. In the context of Ras proteins, this PTM works as a second signal for plasma membrane targeting, after prenylation. The isoform K-Ras4B bypasses the Golgi, and it is not palmitoylated and, therefore, after prenylation travels directly to the plasma membrane [12].

In addition to these CAAX-specific PTM, Ras proteins also undergo other PTM in other amino acid residues along their primary sequence, such as phosphorylation, acetylation, glutathionylation, ubiquitylation and SUMOylation, although the contribution of these PTM appear to be less relevant to the subcellular localization of Ras. Phosphorylation of Ras is the PTM that virtually targets more amino acid residues. The residues Tyr32 and Tyr64 are conserved phosphorylation sites among all Ras isoforms, as well as Tyr137 [19,20,21,22]. In the case of K-Ras4B isoform, Ser181 appear to be the main target of phosphorylation [23]. Other phosphorylation targets include the recently identified Tyr4, Ser89, Thr144 and Thr148, which are also conserved among the four Ras isoforms [24,25,26]. The transfer of an acetyl group from acetyl coenzyme A to lysine residues acetylates Ras proteins. So far, four putative acetylation sites have been described in Ras, Lys101, Lys104, Lys128 and Lys147, with the acetylation of Lys104 and Lys147 being the most well studied [27,28]. Apparently, these PTM do not significantly change Ras function. More recently, it has been found that N-acetylation of Thr2, following deletion of N-terminal Met, is important for the correct folding of Ras and consequently for its function [29]. 

Ras can also undergo ubiquitylation in lysine residues, where the covalent attachment of one or more ubiquitin monomers normally occurs. This PTM can change Ras function by changing the subcellular distribution of the protein or by leading to protein degradation. Ubiquitylation of H-Ras occurs at Lys117, whereas in K-Ras, it occurs mainly at Lys147. The modification of these residues increases Ras activity, although the mechanism underlying each isoform is different. In the case of H-Ras, ubiquitylation increases nucleotide exchange, while in the case of K-Ras, this PTM increases the affinity of Ras for Raf and PI3K [30,31]. 

SUMOylation regards the covalent binding of the small ubiquitin-like modifier (SUMO) protein also to lysine residues. This PTM it is not associated with protein degradation, as in the case of ubiquitylation, and to date, Lys42 has been described as the only target of SUMOylation in all Ras isoforms [32]. 

The highly reactive Cys118 present in Ras proteins is the preferable target for glutathionylation and S-nitrosylation. Glutathionylation is a PTM induced by the action of the tripeptide gluthatione. This PTM occurs mostly in Cys118, but it can also be present in residues Cys80, Cys181, Cys184 and Cys186 [4,33,34]. These post-translational modifications of Ras were recently reviewed by Rhett et al. (2020), for more details refer [1]. 

In this work, we review the S-nitrosylation of Ras and its implications in cellular signaling. S-Nitrosylation is a PTM of cysteine residues, characterized by the nitrosation of the thiol group, which can modify Ras subcellular localization and activity, with several implications on cellular physiology. 

## 4. S-Nitrosylation of Ras

Nitric oxide (NO) is a highly reactive molecule produced enzymatically from L-arginine. In the presence of molecular oxygen and NADPH, three NO synthase (NOS) isoforms catalyze the conversion of L-arginine into L-citrulline with the consequent production of NO. These NOS isoforms encompass the neuronal NOS (nNOS) and the endothelial NOS (eNOS), which are expressed constitutively in neurons and endothelial cells, respectively, and the inducible NOS (iNOS), which is induced in several cell types.

Nitric oxide is classified as a free radical and a gaseous signal transducer with a broad spectrum action within cells. It is well recognized that it can act by two different mechanisms, a classical mechanism or a nonclassical mechanism [35,36]. The classical pathway of NO signaling involves the activation of soluble guanylate cyclase and the production of cGMP, while the nonclassical pathway of NO signaling involves the post-translational modification of proteins by NO and its derivatives by means of S-nitrosylation or S-glutathionylation [35]. S-Glutathionylation, the formation of a disulfide bridge between a protein Cys residue and glutathione, can be independent of NO formation, but it is included as a nonclassical pathway of NO signaling because protein S-nitrosylation or S-nitrosoglutathione (GSNO) formation can be intermediates in the formation of S-glutathionylated products [35,37].

S-Nitrosylation does not occur by directly adding nitric oxide to the thiol of a cysteine residue. Instead, an oxidized species such as N_2_O_3_, produced by the reaction of NO with O_2_ for example, has been described as a significant candidate to modify a reduced thiol group [35,36]. Interestingly, it has been proposed that the interaction of NO with O_2_ can be catalyzed in membranes. For instances, Liu et al. found that in tissues the major fraction of NO reaction with O_2_ occurs at the membrane level instead of occurring at the intracellular milieu [38]. This finding, in addition to the localization of eNOS and nNOS as membrane-bound entities, points to an increased chance of S-nitrosylation in membrane-bound proteins. 

In general, S-nitrosylation is a reversible PTM that occurs when a nitrosyl moiety of NO is covalently incorporated into a thiol group of a cysteine residue in a target protein, resulting in the formation of a S-nitrosothiol (SNO) [35,36]. S-Nitrosylation is one of the best recognized PTM by which NO mediates protein modification in the cellular environment. For instance, S-nitrosylation is a regulatory process of protein conformational change and protein–protein interactions, thus controlling protein function and activity [39]. 

S-Nitrosylation of Ras has been extensively studied and is one of the best examples of how a PTM regulates the signaling of a protein (Figure 2) [6,40,41,42,43,44,45,46]. 

The work of Lander et al. in the 1990’s constituted the first achievements in this field. These authors found that NO activates p21ras in human T cells in a direct and reversible manner, as demonstrated by an increase in GTP-bound p21ras, by means of S-nitrosylation [43]. This process was fundamental to trigger downstream signals such as the activation of NF-kB. Later, it was identified by the same authors that the reaction site of S-nitrosylation was the solvent-accessible Cys118, the most surface-exposed Cys in Ras proteins, in a pioneering example of the detailed characterization of a S-nitrosylation site, both structurally and functionally [47,48]. Cys118 is a conserved residue in all Ras isoforms, close to the active center, and these original studies have demonstrated that modification of specific cysteine residue activates the guanine nucleotide exchange thus providing an alternative way of initiating NO-dependent Ras signal transduction in several contexts.

Very interestingly, Lee and colleagues have found that the expression of iNOS is positively regulated by S-nitrosylation of Ras in two human cell lines (epithelial and astrocyte cells). These authors demonstrated that the mechanism underlying the amplification of NO levels within these cells involves the activation of the PI3K/Akt pathway [49].

## 5. S-Nitrosylation of Ras Enhances Guanine Nucleotide Exchange 

The GTPase activity of Ras proteins can be promoted in the presence of NO/NO-donating reagents, through S-nitrosylation [43,48]. This reaction occurs because NO modifies the most solvent-accessible cysteine, Cys118, at the guanine nucleotide binding site and enhances guanine dissociation [46,48,50] (Figure 2). This results in activation of Ras in vivo and, consequently, contributes to the activation of downstream signaling pathways [6,43,45,47,48,51,52,53]. However, this reaction only modifies Cys118 with no alternative site of Ras modification by NO and does not affect the structure of Ras or effector binding [46,48,50]. 

The link between NO and Ras activation has been shown in different functional contexts. Yun and colleagues showed that activation of N-methyl-D-aspartate (NMDA) receptors promote the activation of nNOS in rodent primary cortical neuronal cultures and that this results in active GTP-bound state of Ras [52]. Additionally, Lin et al. have found that S-nitrosylation of Ras results in the activation of this guanine nucleotide exchanger, and this is something required for the stimulation of the ATP-sensitive potassium channels. These authors concluded that this NO-mediated effect might have a neuroprotective role under ischemic conditions and found that the intracellular signaling underlying this process regards the MAPK cascade rather than the PI3K pathway [54]. Collectively, these data show that S-nitrosylation increases Ras guanine nucleotide exchange. Interestingly, no other oxidative modification can promote this process. Like S-nitrosylation, glutathionylation is an oxidative modification that can modify Cys residues. However, the Cys118 glutathionylation on Ras seems to not affect either Ras structure or nucleotide exchange or GTP hydrolysis. Interestingly, Ras glutathionylation seems to neutralize Ras from further free radical-mediated activation events [55].

More recently, by means of sophisticated computational simulations, it was found that S-nitrosylation of K-Ras affects the dynamics of this protein, specifically at the structural level [56]. These authors demonstrated that S-nitrosylation of K-Ras on Cys118 residue alters the flexibility of the protein leading to a hardening of the Switch I region. This is the region in the structure of the protein where interactions with GAPs and effector molecules occur, meaning that changes in this area have functional implications [56].

## 6. S-Nitrosylation Affects Membrane Association and Subcellular Localization of Ras 

PTMs affect the structure, the function and the localization of proteins within the cell. Recently, the major oxidative modifications targeting the reactive cysteine residues of Ras proteins were extensively reviewed [1,4]. These modifications include S-glutathionylation and S-nitrosylation and determine the way Ras proteins interact with the plasma membrane and with the different subcellular compartments. Ras compartmentalization is triggered by different signaling pathways, which translate into distinct cellular outputs [1,4]. A study aimed to evaluate the interaction behavior of H-Ras with the lipid bilayer in S-nitrosylating conditions (S-nitrosocysteine (CysNO) treatment), by means of a model membrane system (optical waveguide lightmode spectroscopy), found that both cytoplasmic and membrane-anchored H-Ras are modulated by S-nitrosylation (Figure 3A). S-Nitrosylation of the cytoplasmatic Ras inhibits the farnesylation of the protein and its subsequent integration into the plasma membrane, resulting in an enhancement of the GTPase activity (Figure 3A). The cysteines most affected are those located at the C-terminal of the protein (Cys181, Cys184, Cys186). In contrast, S-nitrosylation of the membrane-anchored H-Ras diminishes the rate of GTP hydrolysis (Figure 3A). In this case, S-nitrosylation occurred only at the Cys118 residue, in the GTP-binding pocket [44]. 

Several studies show that the translocation of Ras proteins between the different cellular compartments is regulated by the surrounding oxidative environment. Recent studies have shown that high levels of oxidative stress in hypoxic conditions (high NO levels; simulated with sodium nitroprusside (SNP)) S-nitrosylate H-Ras in its C-terminal cysteines, inducing the translocation of the protein from the plasma membrane to the cytoplasm in PC12 cells [40,57]. These observations took place in undifferentiated but proliferative cells and changed the H-Ras activity. In these conditions, a shift from the Ras/ERK signaling pathway to the PI3K/HIF-1-alpha pathway occurred (Figure 3B). On the contrary, in normoxic conditions (low NO levels), H-Ras remains palmitoylated in its C-terminal cysteines and therefore continues to be associated with the plasma membrane (Figure 3B) [40,57]. These findings are extremely important due to the role of Ras proteins in regulating cellular proliferation and differentiation. Compartmentalization of H-Ras upon S-nitrosylation of Cys118 was also found by Batista and colleagues, with the concomitant activation of different intracellular signaling pathways [41]. An external source of NO (GSNO) induced S-nitrosylation in parallel, for both Src kinase and Ras in HeLa cells. S-Nitrosylation activates the membrane-bound Ras with the subsequent stimulation of the ERK1/2 MAPK pathway. On the other hand, S-nitrosylation of Src by GSNO phosphorylates PLC-γ, ensuing the synthesis of DAG and an increased intracellular Ca^2+^ level. Concomitantly, the translocation of Ras from the cytoplasm to the Golgi occurs, where Ras is activated. This activation of Ras further contributes to the phosphorylation of ERK1/2 (Figure 3C). Conversely, when HUVEC cells are exposed to an endogenous source of NO, such as in the case of eNOS activation following bradykinin-receptor/Akt signaling, the Golgi-associated Ras will not be activated. Instead, NO will S-nitrosylate the plasma membrane-bound Ras, and the ERK1/2 pathway will be stimulated (Figure 3D). Ras activation either with GSNO or eNOS-derived NO promotes proliferation of both HeLa and endothelial cells [41]. A role for eNOS-derived NO upon S-nitrosylation of Ras was also reported by Ibiza et al. [42]. It was found that NO activates N-Ras through S-nitrosylation of Cys118 in T-cells, and this occurs exclusively on the Golgi following T cell receptor stimulation [42,58] (Figure 3E). These findings contrast with those reported by Batista et al. who described that eNOS-derived NO activates the membrane-associated Ras but not the Ras colocalized with Golgi [41]. These divergences can be explained in terms of subcellular localization of the NO source, with eNOS located differentially in the distinct cell types, highlighting the role of S-nitrosylation as a short-range nonclassical NO signaling mechanism [58,59].

A similar mechanism to the one described by Batista et al. for Ras activation in a compartmentalization manner was also observed by Bivona et al., although not involving S-nitrosylation, but instead the stimulation of the EGF receptor [60]. Activation of the EGFR depends on the Src kinase that in turn phosphorylates the PLC-γ and promotes the formation of DAG and an increase in intracellular Ca^2+^ levels. This signaling induces the translocation of Ras to the Golgi where the protein is activated. Ras activation is restricted to this subcellular compartment and has important physiological roles such as T-cell development and function and differentiation of PC12 cells [60]. 

These studies show that S-nitrosylation of Ras affects the membrane association and the subcellular localization of the protein. The compartmentalization of Ras modulates the activity of the protein. The above-mentioned studies also highlight the importance of the NO source regarding the differentiated compartmentalization of Ras. Depending on the stimulus, Ras might be directly activated in a specific subcellular compartment, promoting the stimulation of distinct signaling pathways. This endows the cell with several mechanisms to regulate Ras-associated functions, such as cellular proliferation and differentiation.

## 7. How Is the Interaction between S-Nitrosylation of Ras and Other Targets of S-Nitrosylation in Neuronal Cells?

Ras transduces three main cellular pathways: the MAPK, the PI3K and the RalGEF pathways. Studies have been shown that, like Ras, some of the proteins of the MAPK and PI3K pathways can be activated or inhibited by S-nitrosylation (Figure 4). To date, the S-nitrosylation of elements of the RalGEF pathway has not been described.

### 7.1. MAPK Pathway

Our group demonstrated that S-nitrosylation of Ras is a mechanism independent of the activation of EGF receptor in neural stem cells, meaning that NO bypasses the activation of this receptor [8]. In vivo studies also showed that NO positively contributes to the initial stages of neurogenesis following a hippocampal lesion but compromises survival of newborn neurons [9]. S-Nitrosylation of Cys118 in Ras on NSC increases signaling through the ERK/MAPK pathway by enhancing the phosphorylation of Raf1-MEK-ERK cascade, thus promoting cell proliferation [6,8]. Recently, several S-nitrosylation targets have been identified in NSC and a subset of which are elements of the MAPK pathway, 14-4-3 family, PEBP-1 and hnRNP K [10]. These targets are S-nitrosylated along with Ras. 

It is well established that 14-3-3 proteins are abundantly expressed in the brain and are recognized as promotors of cell proliferation [61]. The members of this family are known to directly interact with b-Raf and c-Raf in order for them to be translocated to the plasma membrane, bind to Ras and initiate the ERK/MAPK signaling [62]. Using a knock-out mice for both 14-3-3 ε/ζ, Toyo-oka and colleagues found that these 14-3-3 isoforms have a main function upon the proliferation and differentiation of neural progenitor cells in the brain cortex of this model [63]. S-Nitrosylation of 14-3-3 ε was also previously described in other cellular contexts, such as in mesangial cells [64] and in primary ovine fetoplacental artery endothelial cells [65], reinforcing 14-3-3 proteins as targets for NO modification. 

PEBP-1 (also known as RKIP, Raf kinase inhibitory protein) has been described in the brain and neuronal tissues [66]. This protein was initially reported in vitro as binding to Raf-1, MEK and ERK, but not to Ras [67]. Presently, in addition of being a direct inhibitor of these MAPK components, it is also described as a regulator of NF-kB cascade, thus controlling several important processes such as proliferation, motility and apoptosis [68]. PEBP-1 is normally regulated by phosphorylation; however, S-nitrosylation of this protein was already described in SH-SY5Y neuronal cells [69]. The involvement of PEBP-1 in the MAPK signaling is well documented in the studies showing that PEBP-1 promotes neuronal differentiation by binding to c-Raf and avoiding MEK phosphorylation, both in a human neuroblastoma cell line [70] and in adult rat hippocampal progenitor cells [71].

The hnRNPs are a family of RNA binding proteins widely expressed by a variety of human cells and with a widespread intracellular location [72]. S-Nitrosylation of hnRNP A/P was previously shown in macrophages cell lines resulting in the inhibition of the activity of these proteins [73]. S-Nitrosylation of hnRNP K was just recently described in NSC (Santos, Lourenco et al., 2020). However, it is well known that the activity of this protein can be regulated by additional PTM such as methylation, ubiquitination, SUMOylation and phosphorylation [72,74]). The inactivation of hnRNP K can regulate the switch from proliferation to differentiation in neural stem cells [75]. It is proposed that hnRNP K can interact with multiple pathways, with ERK1/2 being one of the preferable targets [72,76,77]. 

Overall, 14-3-3, PEBP-1 and hnRNP K have been identified in the brain as targets of NO in NSC cultures and in a murine model of injury-induced neurogenesis [10]. The function of these proteins is mainly related to cell proliferation, and they are pointed as potential candidates for NO-induced proliferation of NSC in a model of brain lesion. As such, we hypothesize that S-nitrosylation of these proteins contributes to proliferation of NSC. The proposed model for this process is depicted on Figure 4. 

### 7.2. PI3K Pathway 

Ras activation can transduce the PI3K signaling pathway. Kwak et al. found that the phosphatase and tensin homolog (PTEN), a regulator of the PI3K/Akt pro-survival pathway with a significant role in neurons, is highly S-nitrosylated in the brain of Alzheimer’s disease patients [78]. Using cultured neuronal cells, these authors found that this PTM signals PTEN for degradation via the ubiquitin-proteasome system (UPS), thus promoting Akt signaling [78]. Also, in the context of neuronal cells, Numajiri et al. demonstrate that PTEN can be S-nitrosylated by low concentrations of NO, which increases Akt signaling, thus contributing to the survival of cells in an ischemic brain. Instead, high concentrations of NO S-nitrosylate Akt resulting in the inhibition of this pathway [79]. Additionally, Choi et al. found that PTEN is inhibited by a similar modification called transnitrosylation (transfer of a NO from the cysteine residue of one protein to another), in neuronal cell lines. The authors also found that this modification decreased PTEN activity thus promoting cell survival [80]. Contrasting to the high number of studies regarding the role of PTEN in cancer, in the neuronal context, these studies are scarce. However, in both contexts, data point to the same hypothesis, i.e., inactivation and consequent degradation of S-nitrosylated PTEN via the UPS. This process releases the inhibitory effect of PTEN and contribute to the survival of cells via Akt signaling. Based on the aforementioned studies, the proposed model for the S-nitrosylation of the PI3K members’ pathway in neuronal cells is the one depicted in Figure 4. 

### 7.3. RalGEF Pathway

The RalGEF pathway is less studied than the MAPK and PI3K pathways, and it is known to be involved in cell division, motility and polarity, among others [81]. Several evidence shows that its activation promotes transformation of human cells [82,83,84]. While the literature does not yet report S-nitrosylation of elements of the RalGEF signaling pathway, several of the members of this pathway present cysteines in its structure, some of which are PTM sites. RalGEF are a family of Ras like guanine nucleotide exchange factors, including RalGPS1 and RalGPS2 isoforms. A cysteine is present at position 341, but PTM have not yet been described for this residue.

Ras like small GTPases RalA and RalB with CAAX carboxy terminal sequences need geranylgeranylated cysteine residues for membrane anchorage [13], which is disrupted by geranylgeranyltransferase I inhibitors (GGTI) [85]. GGTI inhibitors are promising anticancer drugs able to target downstream Ras signaling. It is yet to be determined whether S-nitrosylation of Cys203 in RalA (NP_005393) or RalB (NP_002872), for instance, would interfere with the prenylation of Cys203 by geranylgeranyl transferase activity or with the ability of GGTI inhibitors to disrupt this form of oncogenic signaling. Recently, a nonsense variant Arg176X was identified in a genetic study for de novo mutations of RalA associated with intellectual disability and developmental delay [86]. This variant is predicted to result in a truncated form that lacks the 29 carboxy terminal residues and thus lacks the Cys203 prenylation site, preventing the membrane association of RalA and interfering with the necessary signaling for proper brain development. 

## 8. Conclusions and Future Perspectives

Post-translational modification of Ras by S-nitrosylation affects the membrane association and the subcellular localization of the protein, while also changing its activity. Overall, modification of Cys118 by S-nitrosylation enhances GDP exchanging activity, and thus promotes Ras activation. Several studies have demonstrated S-nitrosylation of C-terminal cysteines in the CAAX may also affect other PTM and change Ras activity and membrane association.

We have addressed the S-nitrosylation of elements in the three main signaling pathways downstream of Ras, which may play a synergistic role on the activation of these pathways. Based on that, we proposed, for the first time, a model for the interaction between S-nitrosylation of Ras and other targets of S-nitrosylation in neuronal cells. Mutations in such elements are further associated in human disease with developmental delay syndromes and cancer. Since S-nitrosylation of several proteins is possible and likely to occur simultaneously due to the nature of NO as a gas transmitter, the further integrated study of such modifications in these pathways should be encouraged. 

Additionally, we have addressed the possibility of source specificity and short-range mechanisms in NO signaling by S-nitrosylation [39,59] in comparison with the view of a generalized alteration of the cellular signaling network by NO. Nevertheless, whether specific NOS cellular sublocalization directs NO to discrete areas or a larger area of the cell or tissue is affected by a bolus of NO produced, for instance, by microglia during neuroinflammation, it should be expected that more than one cellular target is targeted in every signaling cycle. While the analysis of the S-nitrosylation of Ras and its effector pathways is of interest, other signaling pathways affected by S-nitrosylation and players that may be simultaneously altered by NO signaling, including canonical NO signaling mechanisms, should be considered.

## Figures and Tables

**Figure 1 antioxidants-12-01562-f001:**
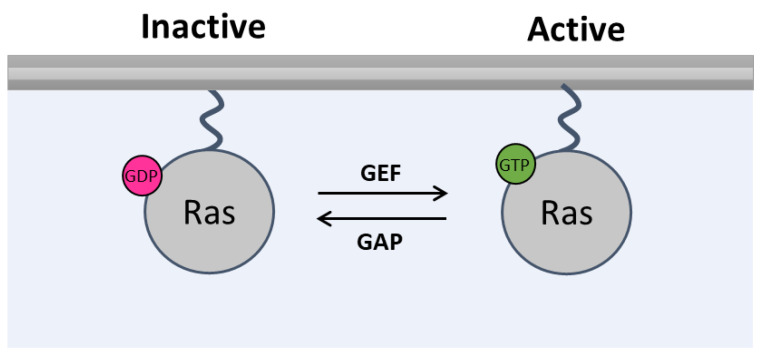
**Canonical activation of Ras.** In the inactive state, Ras is bound to a GDP molecule, and in the active state, Ras is bound to a GTP molecule. This guanine nucleotide exchange is controlled by GEFs and by GAPs. GAPs: GTPase accelerating proteins, GDP: guanosine diphosphate, GEFs: guanine nucleotide exchange factors, and GTP: guanosine triphosphate.

**Figure 2 antioxidants-12-01562-f002:**
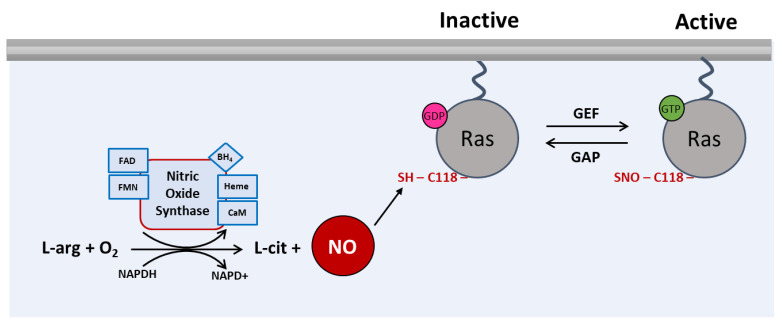
**S-Nitrosylation of Ras.** In the presence of O_2_ and NADPH, three NO synthase (nNOS, eNOS and iNOS) isoforms catalyze the conversion of L-arginine into L-citrulline with the consequent production of NO. The reaction also requires the presence of flavin adenine dinucleotide (FAD), flavin mononucleotide (FMN), tetrahydrobiopterin (BH_4_), heme and calmodulin (CaM). NO reacts with the Cys118 residue of Ras, by means of S-nitrosylation to active the guanine nucleotide exchanger. Arrows (→) in the figure means activation.

**Figure 3 antioxidants-12-01562-f003:**
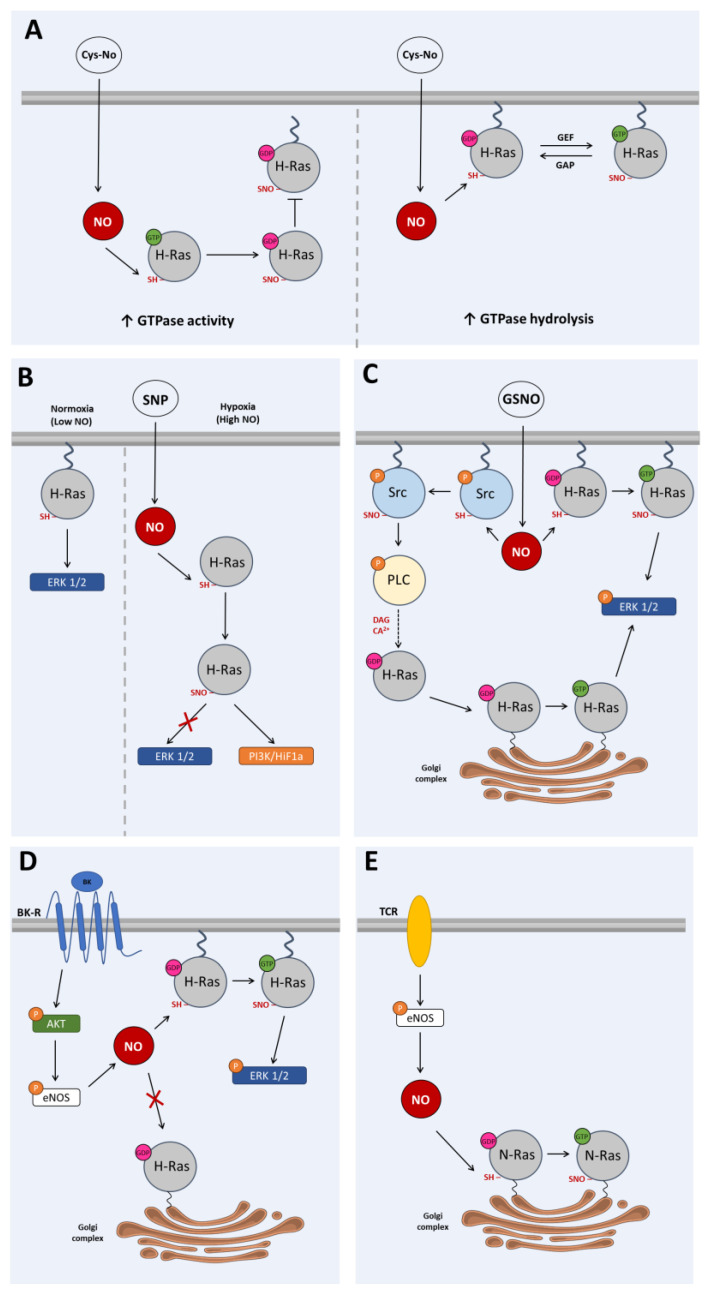
**Changes in subcellular localization of Ras upon S-nitrosylation.** (**A**) S-Nitrosylation of the cytoplasmatic H-Ras by CysNO inhibits the integration of the protein into the plasma membrane, resulting in an enhancement of the GTPase activity. S-Nitrosylation of the membrane-anchored H-Ras by CysNO diminishes the rate of GTP hydrolysis, in a model of membrane system [44]. (**B**) In hypoxic conditions (high NO levels simulated by SNP exposure), Ras is S-nitrosylated with the consequent translocation of the protein from the plasma membrane to the cytoplasm in PC12 cells. In these conditions, a shift from the Ras/ERK signaling pathway to the PI3K/HIF-1-alpha pathway occurred. In normoxic conditions, H-Ras remains associated with the plasma membrane [40,57]. (**C**) Parallel S-nitrosylation of both Src kinase and H-Ras in HeLa cells by means of GSNO. S-Nitrosylation activates the membrane-bound H-Ras with the subsequent stimulation of the ERK1/2 MAPK pathway. S-Nitrosylation of Src phosphorylates PLC-γ, ensuing the synthesis of DAG and an increased intracellular Ca^2+^. The translocation of H-Ras from the cytoplasm to the Golgi occurs, where H-Ras is activated, which further contributes to the phosphorylation of ERK1/2 [41]. (**D**) Activation of the bradykin-receptor/Akt signaling results in eNOS-derived NO production. This translates into S-nitrosylation of the membrane-bound H-Ras and stimulation of the ERK1/2 pathway, in HUVEC cells. In these conditions, the Golgi-associated H-Ras will not be activated [41]. (**E**) Activation of Golgi-associated N-Ras by eNOS-derived NO, following T cell receptor stimulation in T-cells [42,58]. AKT: protein kinase B (PKB), BK: bradykinin, BK-R: bradykinin receptor, CysNO: S-nitrosocysteine, DAG: diacylglycerol, eNOS: endothelial nitric oxide synthase, ERK1/2: extracellular signal-related kinases 1 and 2, GDP: guanosine diphosphate, GSNO: S-nitrosoglutathione, GTP: guanosine triphosphate, HIF1a: hypoxia inducible factor 1 subunit alpha, NO: nitric oxide, PI3K: phosphoinositide-3 kinase, PLC: phospholipase C, SNP: sodium nitroprusside, Src: non-receptor protein–tyrosine kinase (sarcoma), TCR: T cell receptor. Arrows (→) in the figure means activation and red crosses (×) means blockade.

**Figure 4 antioxidants-12-01562-f004:**
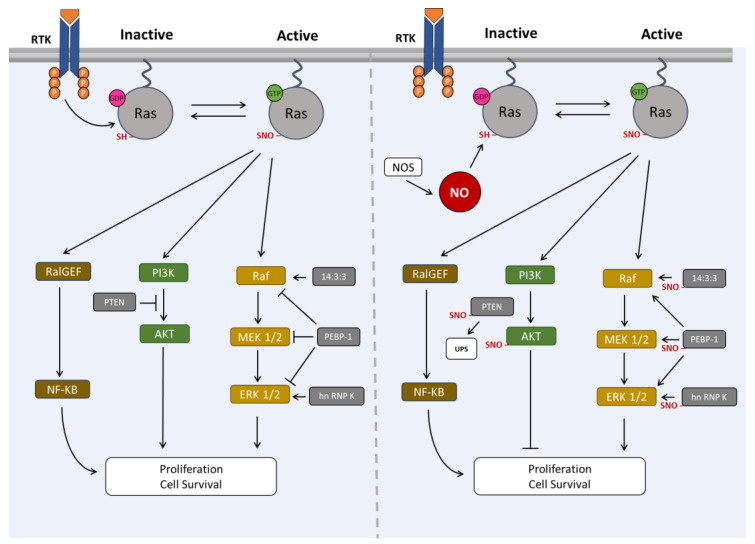
**Proposed model for the interaction between S-nitrosylation of Ras and other targets of S-nitrosylation in neuronal cells**. Ras can be activated following the stimulation of RTK with the consequent transduction of three main signaling pathways: RalGEF, PI3K and MAPK. Ras can also be activated by S-nitrosylation, independent of the stimulation of the RTK. In this case, several members of the PI3K and MAPK cascades are also S-nitrosylated. S-nitrosylation of RalGEF elements in neuronal cells are not known. S-Nitrosylation of 14-3-3 proteins enhances the ERK1/2 MAPK signaling by interacting with the most upstream protein, Raf. S-Nitrosylation of PEBP-1 releases its inhibitory effect upon the Raf/MEK1/2/ERK1/2, promoting the activation of this pathway. hnRNP K S-nitrosylation also contributes to the activation of this MAPK signaling by interacting with ERK1/2. S-Nitrosylated PTEN is signaled to degradation via the UPS, promoting the activation of the Akt cascade. S-Nitrosylation of Akt results in a diminished proliferation of cells. AKT: protein kinase B (PKB), ERK1/2: extracellular signal-related kinases 1 and 2, GDP: guanosine diphosphate, GTP: guanosine triphosphate, hnRNP K: heterogeneous nuclear ribonucleoprotein K, MEK1/2: mitogen-activated protein kinase kinase 1 and 2, NF-kB: nuclear factor kappa B, NO: nitric oxide, NOS: nitric oxide synthase, PEBP-1: phosphatidylethanolamine binding protein 1, PI3K: phosphoinositide-3 kinase, PTEN: phosphatase and tensin homolog, RTK: receptor tyrosine kinase. Arrows (→) in the figure means activation and this symbol (⊥) means blockade.

## Data Availability

Not applicable.

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
