# Peer review of "Regulation of Ras Signaling by S-Nitrosylation"

_antioxidants, 2023, doi:10.3390/antiox12081562_

Round 1

Reviewer 1 Report

In the paper titled Regulation of Ras signaling by S-nitrosylation the authors affirm to present a better understanding of Ras S-nitrosylation, allowing the knowledge of the dynamics of this protein in cellular signaling.

Being an updating review, I would invite the authors to use a more up-to-date bibliography. It is not possible to enter entries that are more than thirty years old.

Furthermore, where the authors refer to experimental trials, indicate the corresponding reference, avoiding inserting a review in this context.

Page 4 line 146. This is a redundant sentence, just reported above in the manuscript. 

Apart from this sentence, many concepts are redundant in the body of the manuscript. Apart from this sentence, many concepts are redundant in the body of the manuscript.

This paper does not seem to bring anything particularly interesting in addition to what is already known in the literature.

Moderate editing of English language required

Author Response

Please see the attachment. A pdf file was uploaded.

Reviewer 2 Report

In this review, the mechanisms of regulation of a wide range of RAS-related signals by nitrosylation are described, which I thought would be a good suggestion for the field. On the other hand, the structure is somewhat selective for the reader, as there are scattered instances where technical terms are used without explanation. I thought it should be written in an explanatory manner with a broader audience in mind. I hope that authors will find the following points helpful.

 I found it difficult to understand the dizzying change of topics with each section talking about nerves and cancer. why not explain the physiological functions of the RAS with cancer/neurology in a summary first? 

 Lines 113-115 should be accompanied by a citation or specific explanation.

 RAS PTMs are diverse. It may not be necessary to explain them all, but we felt that the most important PTMs should be explained. Since this is a debut for a diverse audience, I felt it was necessary to at least explain the process by which NO is released and the differences between PTMs caused by NO and other PTMs.

 I'm having trouble understanding the connection between the section on lines 160-166: where is the similarity between NMDA-induced nNOS-induced RAS activation and tumorigenic potential? Maybe a more specific explanation is needed. I did not understand what they were trying to explain.

 Lines 321-326 are unclear because there is no conclusion. how does SNO affect hnRNP?

 The descriptions related to Figure 2 are quite fragmentary and difficult to understand, so it would be better to cite the figure where A~E have relevant descriptions. Also, what does Cys-NO mean in Figure 2A?

 Also, there seems to be a description that only exists in the figure. For example, is the SNP in Figure 2B mentioned in the text? I find this a bit unfriendly as a review. If the figures and the explanations corresponded on a one-to-one basis, this kind of part would be eliminated.

 Figure 3 should be a diagram that shows a little more about the correlation between the proteins that appear. At first glance, the content is not clear.

 Lines 419-421 are written in a way that only researchers in the field would understand. Similar problems can be found elsewhere, making this review quite technical. We recommend that it be written in a more descriptive manner with a wider readership in mind. The structure is also somewhat difficult to follow, so I personally recommend that someone not familiar with the field read it and identify the problems.

Author Response

(The authors gave the same response as above.)

Reviewer 3 Report

The manuscript (MS) entitled “Regulation of Ras signaling by S-nitrosylation” by Sónia Simão and colleagues presents a review paper discussing posttranslational RAS protein modification and its impact on cellular signalling, with focus on S-nitrosylation. The MS is concise, informative, and well written. It provides a concise summary of what is known and defines where the lack of evidence is present.

Minor points:

1. Partially due to the topic and “cellular” nomenclature, the MS is overloaded with abbreviations,  which may interfere with ease of reading. I suggest addition of list of abbreviations used in the MS with their explanations.

2. The MS will benefit from careful proofreading and some minor editing corrections. Here are some examples, please, note there are more issues in the MS requiring correction.

L90-91:  please, provide reference for Rhett et al. (2020). Now, it is number 51, but should be introduced at this point. Please, note that numbering of references will require update.

L169: “consists on” – probably should be “consists of”

L296-297:  “Studies have 296 been shown that” – should be “Studies have shown that”

3. In my view, authors could reference some recent review papers on RAS protein, which give a broader context on biology and therapeutic context:

Campbell SL, Philips MR. Post-translational modification of RAS proteins. Curr Opin Struct Biol. 2021 Dec;71:180-192. doi: 10.1016/j.sbi.2021.06.015. Epub 2021 Aug 6. PMID: 34365229; PMCID: PMC8649064.

Mukhopadhyay S, Vander Heiden MG, McCormick F. The Metabolic Landscape of RAS-Driven Cancers from biology to therapy. Nat Cancer. 2021 Mar;2(3):271-283. doi: 10.1038/s43018-021-00184-x. Epub 2021 Mar 24. PMID: 33870211; PMCID: PMC8045781.

(please, note, I am note affiliated with any of the papers).

Author Response

(The authors gave the same response as above.)

Round 2

Reviewer 1 Report

No comment

Reviewer 2 Report

I think the authors did a good job of responding to the reviewers' requests. This version has a clear flow of logic and easy to understand figures.

I have no additional remarks to make.